# SH3BP2 Silencing Increases miRNAs Targeting ETV1 and Microphthalmia-Associated Transcription Factor, Decreasing the Proliferation of Gastrointestinal Stromal Tumors

**DOI:** 10.3390/cancers14246198

**Published:** 2022-12-15

**Authors:** Elizabeth Proaño-Pérez, Eva Serrano-Candelas, Cindy Mancia, Arnau Navinés-Ferrer, Mario Guerrero, Margarita Martin

**Affiliations:** 1Biochemistry and Molecular Biology Unit, Biomedicine Department, Faculty of Medicine and Health Sciences, University of Barcelona, 08036 Barcelona, Spain; 2Clinical and Experimental Respiratory Immunoallergy (IRCE), Institut d’Investigacions Biomediques August Pi i Sunyer (IDIBAPS), 08036 Barcelona, Spain; 3Faculty of Health Sciences, Technical University of Ambato, Ambato 180105, Ecuador

**Keywords:** SH3BP2, MITF, ETV1, miRNA, cell survival, cell cycle, gastrointestinal stromal tumors

## Abstract

**Simple Summary:**

A previous study showed that silencing the adaptor molecule SH3 Binding Protein 2 (SH3BP2) reduced oncogenic KIT and PDGFRA receptor levels and impaired gastrointestinal stromal tumor (GIST) growth. This study tries to get insights into the molecular mechanism underlying this effect. The silencing of SH3BP2 induces miRNAs (miR-1246 and miR-5100), which target microphthalmia-associated transcription factor (MITF) and ETV1, a linage survival factor involved in GIST tumorigenesis. Altogether, this results in decreased tumor cell viability and enhanced apoptosis.

**Abstract:**

Gastrointestinal stromal tumors (GISTs) are the most common mesenchymal tumors of the gastrointestinal tract. Gain of function in receptor tyrosine kinases type III, KIT, or PDGFRA drives the majority of GIST. Previously, our group reported that silencing of the adaptor molecule SH3 Binding Protein 2 (SH3BP2) downregulated KIT and PDGFRA and microphthalmia-associated transcription factor (MITF) levels and reduced tumor growth. This study shows that SH3BP2 silencing also decreases levels of ETV1, a required factor for GIST growth. To dissect the SH3BP2 pathway in GIST cells, we performed a miRNA array in SH3BP2-silenced GIST cell lines. Among the most up-regulated miRNAs, we found miR-1246 and miR-5100 to be predicted to target *MITF* and *ETV1*. Overexpression of these miRNAs led to a decrease in MITF and ETV1 levels. In this context, cell viability and cell cycle progression were affected, and a reduction in BCL2 and CDK2 was observed. Interestingly, overexpression of MITF enhanced cell proliferation and significantly rescued the viability of miRNA-transduced cells. Altogether, the KIT-SH3BP2-MITF/ETV1 pathway deserves to be considered in GIST cell survival and proliferation.

## 1. Introduction

Gastrointestinal stromal tumors (GISTs) are the most common type of soft tissue sarcoma in the intestinal tract [1]. They are derived from the interstitial cells of Cajal (ICCs), located in the submucosa and myenteric plexus of the gastrointestinal tract [2]. The pathogenesis of GISTs is defined by mutually exclusive mutations in *KIT* (75–80%) and platelet-derived growth factor receptor α (*PDGFRA*) genes (5–10%). Additionally, 10–15% of GISTs lack *KIT* mutations, the so-called “wild type.” They are classified as deficient in succinate dehydrogenase (SDH)-deficient and non-SHD-deficient. Non-SDH-deficient include NF type 1 neurofibromatosis and GISTs with *BRAF*, *KRAS*, and *PIK3CA* mutations [3].

SH3BP2 (cytoplasmic adaptor molecule SH3-binding protein 2) has been described as an active regulator of *KIT* expression and signaling in mast cells [4] and GIST cells [5]. The silencing of SH3BP2 decreases KIT levels and increases the caspase-3/7 activity, which consequently induces apoptosis. Additionally, SH3BP2 regulates KIT at the transcriptional level and MITF (microphthalmia-associated transcription factor) at the post-transcriptional level in mast cells [4]. Overexpression of MITF in GIST cell lines prevented significant cellular apoptosis [5]. MITF is a basic helix-loop-helix leucine zipper, a dimeric transcription factor well-documented in melanocyte differentiation, cell cycle progression, and survival by targeting pigment enzyme genes or CDK2 [6,7], among others. MITF activity is needed for melanocyte development, and deregulation of its activity is reported in melanoma [8]. Besides melanocytes, MITF is essential for mast cell differentiation [9] and binds to the KIT promoter on mast cells [10]. MITF has recently been reported to be involved in GIST cell survival, proliferation, and tumor growth, and MITF silencing leads to an ETV1 reduction [11]. ETV1 is a transcription factor required for the development of interstitial cells of Cajal and the proliferation of GIST cells [12]. ETV1 is regulated by the MEK–MAPK pathway downstream and activated by KIT and PDGFRA [13]. Therefore, KIT inhibition with imatinib reduces ETV1 levels [14]. Likewise, we found that GIST treated with imatinib reduced MITF levels in vitro [5].

Due to the regulation of SH3BP2 over KIT and MITF levels and their mutual regulation, herein, we aimed to study ETV1 involvement in the pathway and tried to dissect the SH3BP2 pathway pursuing the analysis of miRNAs. In this study, a miRNA microarray was performed, comparing the expression levels of several miRNAs in SH3BP2-silenced GIST cell lines with non-silenced cells. We analyzed the highest up-regulated miRNAs that predictively regulate MITF or ETV1 levels. Further, these miRNAs were validated and characterized in GIST cell lines.

## 2. Materials and Methods

### 2.1. Antibodies and Reagents

Mouse anti-SH3BP2 (clone C5), mouse anti-KIT (clone Ab81), mouse anti-BCL2, and mouse anti-CDK2 were purchased from Santa Cruz Biotechnology, Inc. (Santa Cruz, CA, USA). Anti-MITF (clone D5G7V) was obtained from Cell Signaling Technology, Inc (Danvers, MA, USA). Mouse anti-β-actin (clone AC-40) was purchased from Sigma (St. Louis, MO, USA). Anti-ETV1 antibody (ER81) (ab81086) was obtained from Abcam technology (Abcam, Cambridge, UK). Anti-mouse and anti-rabbit IgG peroxidase Abs were acquired from DAKO (Carpinteria, CA, USA) and Biorad (Hercules, CA, USA), respectively.

### 2.2. Cell Culture

Human GIST cell lines GIST882, GIST48, and GIST-T1 were kindly provided by Dr. S. Bauer. GIST cell lines were-cultured as described elsewhere [11]. Transient transfections were carried out using Opti-MEM (Gibco, Carlsbad, CA, USA). The mycoplasma test was performed routinely in all cell lines used.

### 2.3. RNA Extraction, Retrotranscription, and PCR Assays

Total RNA was extracted with a miRCURY RNA Isolation Kit (Exiqon, Vedbaek, Denmark) from NT control and SH3BP2 knockdown GIST cells. cDNA was generated by reverse transcription using the miRCURY LNA RT Kit. Quantitative, Real-Time PCR for miRNA PCR assay was performed using the miRCURY SYBR Green PCR Kit, and following miRCURY LNA miRNA PCR assay protocol on a LightCycler^®^ 480 Instrument II (LifeScience Roche). miR-30c-5p and miR-335 were used as housekeeping miRNA genes.

### 2.4. MicroRNA Array Profiling

All experiments were conducted at Exiqon Services, Denmark. The quality of all the total RNA was verified by an Agilent 2100 Bioanalyzer profile. 750 ng total RNA from both sample and reference was labeled with Hy3™ and Hy5™ fluorescent labels, respectively, using the miRCURY LNA™ microRNA Hi-Power Labeling Kit, Hy3™/Hy5™ (Exiqon, Vedbæk, Denmark), following the procedure described by the manufacturer. The Hy3™-labeled samples and a Hy5™-labeled reference RNA sample were mixed pair-wise and hybridized to the miRCURY LNA™ microRNA Array 7th Gen (Exiqon, Denmark), which contains capture probes targeting all microRNAs for humans, mice, or rats registered in the miRBASE 18.0. The hybridization was performed according to the miRCURY LNA™ microRNA Array Instruction manual using a Tecan HS4800™ hybridization station (Tecan, Austria). After hybridization, the microarray slides were scanned and stored in an ozone-free environment (ozone level below 2.0 ppb) to prevent potential bleaching of the fluorescent dyes. The miRCURY LNA™ microRNA Array slides were scanned using the Agilent G2565BA Microarray Scanner System (Agilent Technologies, Inc., Santa Clara, CA, USA), and the image analysis was carried out using the ImaGene^®^ 9 (miRCURY LNA™ microRNA Array Analysis Software, Exiqon, Denmark). The quantified signals were background corrected (Normexp with offset value 10, see [15]) and normalized using the global Lowess (Locally Weighted Scatterplot Smoothing) regression algorithm. Among the 502 human miRNAs detected by the array, we discarded any miRNA with an Average Hy3 signal under 7.5.

### 2.5. Lentiviral Transduction

Lentiviral particles to silence the *SH3BP2* gene expression were previously described [5]. Lentiviral transduction for NT (non-target) was performed as described in [4] with slight modifications. PLenti-III-mir-GFP-blank was the plasmid used as a control. Plenti-III-miR-GFP miRNAs (miR-1246 and miR-5100) were obtained from Applied Biological Materials Inc (Richmond, BC, Canada). GIST cells were transduced in the presence of 8 μL/mL of Polybrene (Santa Cruz, CA, USA), and puromycin selection (1 μg/mL) was carried out after one day from transduction.

### 2.6. Cell Viability, Proliferation, and Caspases 3/7 Activity Assays

Cell viability and proliferation were assessed using Crystal violet dye [16], colorimetric assay (WST-1 based) (Roche Diagnostics, Germany), and caspase activity using the Caspase-Glo™ 3/7 Assay (Promega, San Luis Obispo, CA, USA), according to the manufacturer’s protocol.

### 2.7. Western Blotting

Western blotting was performed as described [5,17]. Briefly, transduced cells were lysed at the 5th and 7th days post-lentiviral infection. Electrophoresis and protein blotting was performed using NuPage TM 4–12% Bis-Tris Gel, 1.5 mm × 15 w (Invitrogen, Waltham, MA, USA), and electrotransferred to polyvinylidene difluoride (PVDF) membranes (Millipore, Bedford, MA, USA). Blots were probed with the indicated antibodies. In all blots, proteins were visualized by enhanced chemiluminescence (WesternBright TM ECL, Advansta, San Jose, CA, USA). Original blots see Appendix A.

### 2.8. Cell Cycle Analysis by Flow Cytometry

GIST cells were collected on the 5th and 7th days after transduction. The cells were fixed with 70% ethanol at 4 °C overnight and stained with propidium iodide buffer, as described elsewhere [18]. Data were acquired in FACS Calibur and analyzed using model Dean/Jet/Fox FlowJo 7.6 software.

### 2.9. MITF Overexpression

MiR-CTL, miR-1246, and miR-5100 were overexpressed by lentiviral transduction in GIST-T1. Cells were selected with puromycin (1 μg/mL) 24 h after infection. MITF overexpression was achieved using MITF A GFPpcDNA 3.1+/C-eGFP or pEGFP N3 (control) plasmid (Genscript). MITF-GFP or GFP plasmids were transfected into GIST-T1 cells by Lipofectamine LTX (Invitrogen) on the 3rd day after transduction following the manufacturer’s instructions with slight modifications. Mix plasmids and lipofectamine were incubated overnight with CTS Opti-MEM (Gibco). Cells were maintained in IMDM media (Lonza) with puromycin (1 μg/mL). Cell proliferation was determined on the 7th day using WST-1 (Roche Diagnostics, Mannheim, Germany). MITF and MITF-GFP protein levels were analyzed by Western blotting.

### 2.10. LNA Anti-miRNA Treatment

GIST-T1 (0.06 *×* 106 cells/w) was cultured in a 96-well plate and treated gymnotically with fluoresceinated LNA oligonucleotides (LNA^®^, miRCURY^®^, QIAGEN, Hilden, Germany). LNA anti-miR-1246 (0.1 µM), LNA anti-miR-5100 (0.1 µM), and LNA mixed solution (anti-miR-1246, 0.05 µM + anti-miR-5100, 0.05 µM). Two hours after LNA treatment, sh3BP2 lentiviral particles were added with 8 µg/mL polybrene (Santa Cruz). GIST cells were selected by puromycin (1 μg/mL) after 24 h of lentiviral transduction. We measured cell viability by crystal violet assay [16] on the 4th day post-transduction.

### 2.11. Statistical Data Analysis

After determining the normal distribution of the samples and variance analysis, an unpaired student’s *t*-test was used to determine significant differences (*p*-value) between the two experimental groups. A one-way ANOVA test was used to determine significant differences (*p*-value) between several experimental groups. All results are expressed as mean ± standard error of the mean (SEM).

## 3. Results

### 3.1. SH3BP2 Silencing Reduces ETV1 Levels in GIST Cell Lines

In previous work, we showed that silencing of SH3BP2 diminished KIT, PDGFRA, and MITF levels lead to a reduction in tumor growth in vitro and in vivo [5]. To better understand the role of SH3BP2 in GIST survival, we checked whether silencing of SH3BP2 was also affecting ETV1, a master regulator of the normal linage of interstitial Cajal cells, which cooperates with KIT in GIST [14,19]. As shown in Figure 1 and Appendix A, silencing of SH3BP2 reduces MITF, as previously reported, and ETV1 protein levels in imatinib-sensitive and imatinib-resistant GIST cell lines.

### 3.2. miRNA Profiling of SH3BP2-Silenced GIST Cells

SH3BP2 silencing reduces MITF at the protein level but not at the mRNA level in GIST [5], suggesting a post-transcriptional regulatory mechanism. KIT can regulate MITF through selective miRNA expression in mast cells [12]. Thus, we next performed a miRNA microarray to identify miRNAs regulated by SH3BP2 in GIST882 and GIST48-silenced cells to get insights into the signaling pathway that leads to apoptosis. Figure 2 shows the heat map representation of the two-way hierarchical clustering of miRNAs and samples. Interestingly, the samples cluster according to their biological group, meaning a very different miRNA profile exists between Non-Target and SH3BP2-silenced cells independently of the cell type.

A *p*-value < 0.05 was used to define significantly deregulated miRNAs between the different groups. This criterion identified 162 and 130 miRNAs in GIST882 and GIST48, respectively, with 107 in common (Figure 2A). In Figure 2B, a four-way Venn diagram shows that 32 miRNAs are downregulated in both cell lines, and 56 are up-regulated among the significantly changed miRNAs. Among them, a threshold of 1.5-fold change defined the 21 most up-regulated and the 12 most downregulated miRNAs in both cell lines (Figure 2C). Several databases were used to predict miRNA–target interactions with these miRNAs (Appendix A).

### 3.3. Validation of Up-Regulated miRNAs That Target MITF and ETV1in GIST Cell Lines

From the most up-regulated miRNAs, we identified microRNAs that target MITF and ETV1. We used TargetScan [20], miRtar [21], miRwalk 2.0 [22], microT CDS [23], and mirDIP [24]. The different databases identified miR-1246, miR-1264, miR-1290, miR-3182, and miR5100 as putative MITF and ETV1 partners [25]. The results are summarized in Appendix A.

Next, we validated these five miRNAs in various GIST cell lines. Quantitative real-time PCR was carried out in SH3BP2 silenced GIST-T1 (Figure 3A), GIST882 (Figure 3B), and GIST48 cells (Figure 3C). Only two of the five putative miRNAs (miR-1246 and miR-5100) exhibited significant differences between SH3BP2 shRNA and scramble transfection in all GIST cells. In parallel, only miR-1246 and 5100 overexpression in GIST cell lines show a reduction of MITF level by western blot (Appendix A). We restricted further studies to these two miRNAs. The miRNAs sequence location on the *MITF-A,* the highest isoform expressed, and ETV1 genes, are shown in Appendix A.

### 3.4. MiR-1246 and miR-5100 Target ETV1 and MITF, and Overexpression Significantly Affects Cell Proliferation

As mentioned above, these miRNAs putatively bind to *MITF* or *ETV1* mRNA, so we overexpressed them in the imatinib-sensitive GIST-T1 and imatinib-resistant GIST-48 cell lines to check ETV1 and MITF protein levels. The overexpression of GFP-miR-1246 and GFP-miR-5100 efficiently causes the downregulation of MITF and ETV1 protein levels (Figure 4A,B and Appendix A). Consistently with this, we reported diminished cell proliferation in GIST cells, Figure 4C. The levels of transfection were similar in all cases (Appendix A).

### 3.5. MiR-1246 and miR-5100 Promote Apoptosis by Caspases 3/7 in GIST Cells

To analyze how miRNAs affect cell proliferation, we performed a viability assay and measured caspase 3/7 activity on overexpressed miRNAs GIST cells. Our results show a decrease in cell viability that correlates with an increase in caspase 3/7 activity in both cell lines (Figure 5B,C). Previous studies reported that miR-5100 induces apoptosis throughout caspase 3 protein activity [26], and miR-1246 increases apoptosis by promoting caspase 3 and caspase 7 activity [27]; these results are consistent with the anti-apoptotic protein BCL2 (MITF-dependent target) reduction after overexpression of miRNAs (Figure 5A and Appendix A).

### 3.6. MiR-1246 and miR-5100 Affect Cell Cycle Progression

MITF regulates CDK2 in melanoma, which is critical for tumor cell growth [7,28]. We further analyzed whether CDK2 was altered after miRNA overexpression. MITF reduction was accompanied by decreased CDK2 levels in GIST-T1 and GIST 48 (Figure 6A,B and Appendix A). The overexpression of both miRNAs had different consequences in the cell cycle in both cell lines. GIST-T1 (Figure 6C) overexpression induced a substantial increase in the G2 phase, while in GIST 48 (Figure 6D), there is an accumulation in the S phase. Altogether, these results indicate that these miRNAs may regulate MITF-dependent targets and cell cycle progression.

### 3.7. MITF Overexpression Significantly Restores Cell Proliferation

Next, we assessed the specificity of the effect of MITF on the proliferation of miRNA-treated GIST cells. For that purpose, after three days of miRNA transduction (when cells were still viable), cells were transfected with MITF-GFP or GFP. Seven days after miRNA transduction, MITF levels and cell proliferation were assessed. Our data show that MITF reconstitution is detected by western blot (Figure 7A and Appendix A) and significantly increases cell proliferation (Figure 7B).

### 3.8. LNA Treatment (Anti-miR-1246 Anti-miR-5100) Was Not Adequate to Revert the Apoptotic Phenotype in GIST-T1 SH3BP2 Silenced Cell

To determine whether miR-1246 and miR-5100 are the main ones responsible for the SH3BP2 silencing apoptotic phenotype, we analyzed the effect of LNA (Locked nucleic acid) miRNA inhibitors treatment on SH3BP2 silenced cells. LNA miRNA inhibitors are antisense oligonucleotides with perfect sequences complementary to their target miRNA that prevent miRNA hybridization with its regular cellular interaction partners. LNAs are taken up naturally by cells by a process known as gymnosis. We checked if LNA treatment reverted the apoptotic phenotype of SH3BP2 silenced cells. For that purpose, GIST-T1 cells were treated with LNA against miR-1246 and miR-5100, and afterward, cells were transduced with lentiviral particles shRNA-SH3BP2. Effective gymnosis was measured by FAN fluorescence microscopy each 24 h in treated cells. Our data show that LNAs treatment cannot block the apoptotic phenotype promoted by SH3BP2 silencing (Appendix A). These results suggest that SH3BP2 action on apoptotic phenotype goes beyond miR-1246 and miR-5100.

## 4. Discussion

GISTs can be successfully treated with imatinib or other TKIs [1,29]. However, the necessity for new therapeutical approaches arose due to clinical resistance. We previously reported that silencing of SH3BP2 leads to a reduction of *KIT* expression at both mRNA and protein levels, as well as MITF at the protein level, resulting in a decrease in GIST tumor growth in vitro and in vivo [5]. In the same study, overexpression of MITF significantly reverses the apoptotic phenotype produced by SH3BP2 silencing, suggesting the involvement of this transcription factor in the regulatory mechanism in which SH3BP2 levels are critical. SH3BP2 silencing did not alter *MITF* mRNA levels but protein levels, suggesting a post-transcriptional mechanism. A miRNA microarray was performed in SH3BP2-silenced GIST882 and GIST48 cell lines (imatinib-sensitive and resistant cells) compared to non-silenced cells to get insights into the KIT-SH3BP2-MITF pathway. This microarray showed a different miRNA pattern when SH3BP2 was silenced. In parallel, we found that SH3BP2 silencing also targets ETV1, a master of ICC-transcription factor whose regulation is dependent on KIT signaling and is directly involved in the tumorigenic phenotype [14,19]. In this study, from the top miRNAs that were up-regulated, we focused on those that putatively target MITF and ETV1. After database analysis and cell validation, the miRNAs: miR-1246 and miR-5100 were selected for further studies. Overexpression assays showed that these miRNAs targeted MITF and ETV1 in GIST48 and GIST-T1. Consequently, the decrease in the levels of these transcription factors leads to a reduction in cell survival.

In this context, miR-1246 has been described as a tumor suppressor miRNA in prostate cancer, as authors showed that miR-1246 overexpression led to the inhibition of xenograft tumor growth over time [30]. They propose the exosomal-mediated release of miR-1246 to serum from tumor cells to evade its tumor suppressor role. Moreover, they suggest exosomal miR-1246 as a good biomarker to discern between benign of aggressive prostate cancer. Interestingly, exosomal miR-1246 has been proposed as a biomarker in gastric cancer (GC), and bioinformatics analysis revealed it as a tumor suppressor in GC [31]. Moreover, miR-1246, which can be induced by tumor suppressor p53, has been described as a tumor suppressor due to its capacity to reduce DYRK1A (a Down syndrome-associate kinase) levels, leading to the nuclear retention of NFATc1 and the induction of apoptosis [32]. Additionally, miR-1246 was downregulated in thyroid cancer, and the overexpression of miR-1246 affects PI3K/AKT pathway by regulating phosphoinositide 3-kinase adapter protein1 (PIK3AP1), resulting in less cell proliferation, diminished migration, and increasing apoptosis [33]. Furthermore, miR-1246 mediates LPS-induced pulmonary endothelial cell apoptosis in vitro and acute lung injury (ALI) in mouse models, which are at least partly attributed to the suppression of angiotensin-converting enzyme 2 (ACE2) [27]. In addition, miRNA-1246 mediates ALI-induced lung inflammation and apoptosis via the NF-κB activation and Wnt/β-catenin suppression [34]. Additionally, miRNA-1246 attenuates renal cell carcinoma’s proliferative and migratory abilities by downregulating CXCR4 [35]. Nonetheless, the oncogenic role of miR-1246 has been reported in melanoma by conferring resistance to BRAF inhibitors [36] or enhancing migration and invasion through the adhesion molecule CADM1 in hepatocellular cancer [37].

Regarding miR-5100 activity as a tumor suppressor, our results follow Chijiiwa et al. [38]. The authors show that miR-5100 decreases the aggressiveness of the pancreatic cancer tumor models through the inhibition of PODXL, which promotes anti-adhesive and migratory characteristics of various cancers, and high levels of PODXL correlates with poor prognosis in many of them. Moreover, miR-5100 can increase the apoptosis level of gastric cancer cells and inhibit autophagy by targeting CAAP1 (conserved anti-apoptotic protein 1 or caspase activity and apoptosis inhibitor 1) [26].

However, miR-1246 and miR-5100 have been reported as oncogenic miRNAs in lung cancer [39]. One explanation for these contradictory results could be that miRNAs may vary their affinity to target mRNA depending on the cell lines, the pool of miRNAs that they could be cooperating, and the secondary structures in the 3′UTR of the target mRNA, which can affect the binding of a miRNA [40]. In conclusion, many factors could interfere with the miRNA functional effect in other cell lines.

The proapoptotic role of miR-1246 and miR-5100 in GIST cell lines could result from their ability to affect the cell cycle and regulate cell apoptosis. These actions can be related to a MITF reduction since BCL2 and CDK2 are MITF-dependent targets [7,41,42,43]. BCL2 is found in most GIST patients [19] and correlates with a poor prognosis before imatinib treatment [44]. These miRNAs also induce cell cycle arrest in a cell line-dependent manner. CDK2 has been reported to regulate both G1/S and G2/M transitions. [45]. As previously noted, high double-negative CDK2-expressing cells were arrested in the mid-S phase. In contrast, low double negative CDK2 expressing cells progressed through early and mid-S phases but were still arrested in the late S/G2 phase [45], suggesting that the active CDK2 can be critical in the different phases. Recent research has shown that CDK2 deficiency slows colorectal cancer’s S/G2 progression [46]. It would deserve further consideration to analyze the role and regulation of CDK2 in the different GIST cell lines.

The blockage of miR-5100 and 1246 using LNA did not reduce apoptotic effects due to SH3BP2 silencing, indicating that other miRNAs contribute to this phenotype. However, overexpression of MITF significantly restores cell survival after miR-5100 and 1246 transduction, suggesting that MITF is a crucial target for cell viability.

The role of MITF is well-known in melanoma [47], and recent studies suggest that MITF overexpression in kidney angiomyolipoma cells [48] and clear cell renal cell carcinoma (ccRCC) improve cell growth, proliferation, and invasion in vitro and in vivo [49]. Lately, we have described that the silencing of MITF results in decreased gastrointestinal stromal tumor cell viability in vitro and tumor growth in vivo [11].

## 5. Conclusions

Our results highlight the KIT-SH3BP2-MITF/ETV1 pathway for GIST cell survival and proliferation. Targeting ETV1 and MITF together will help break the positive feedback loop and indirectly target KIT independently of the mutations in the tyrosine kinase receptor.

## Figures and Tables

**Figure 1 cancers-14-06198-f001:**
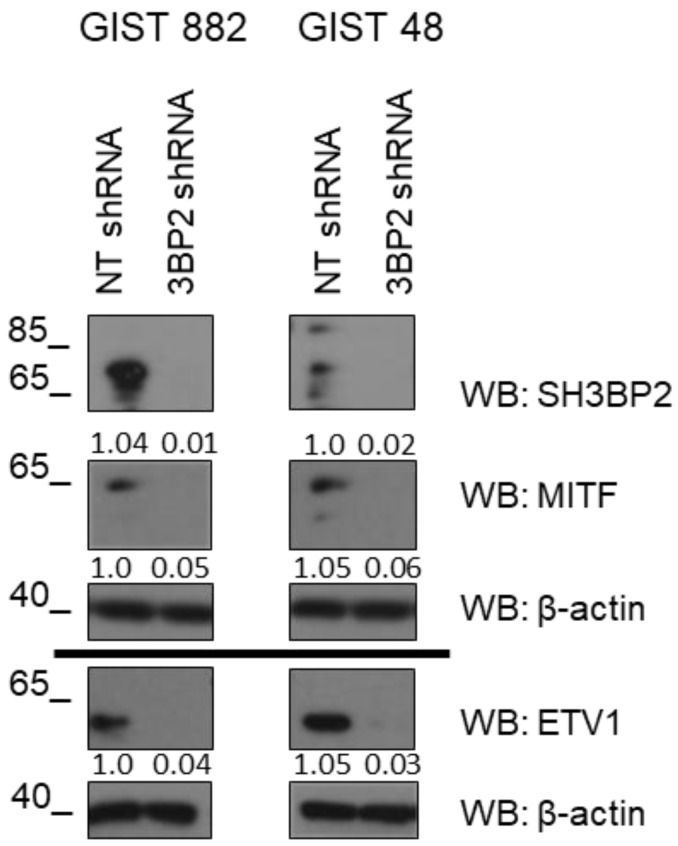
Reduced levels of ETV1 and MITF in SH3BP2-silenced GIST cells. GIST882 and GIST48 were transduced with control NT (Non-target) shRNA and SH3BP2 shRNA. Cell lysates were analyzed on the 7th day post-transduction for MITF, ETV1, and SH3BP2. β-actin was used as a loading control.

**Figure 2 cancers-14-06198-f002:**
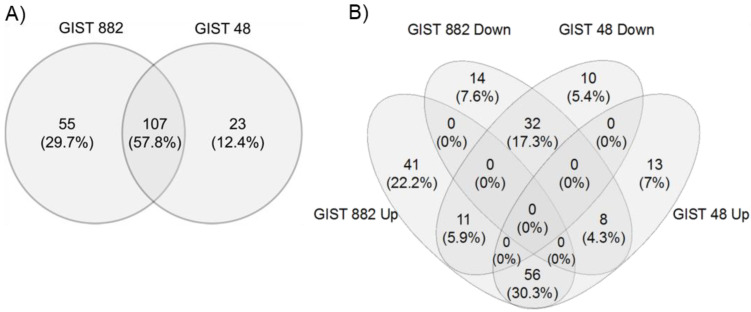
Profile of miRNA expression in SH3BP2-silenced GIST cells. Heat map representation of two-way hierarchical clustering of the miRNAs altered in GIST48 and GIST882 after 3BP2 silencing. (**A**) Venn diagram of miRNAs altered after SH3BP2 silencing. The expression of 107 miRNAs was significantly altered in both cell lines (*p*-value < 0.05). (**B**) A four-way Venn diagram shows the overlapping of the different miRNAs in both cell lines. The clustering was done using the complete-linkage method and Euclidean distance measure. (**C**) Each column represents a single sample, and each file represents a single miRNA. The red and blue colors represent high and low relative expressions, respectively (*p*-value < 0.05).

**Figure 3 cancers-14-06198-f003:**
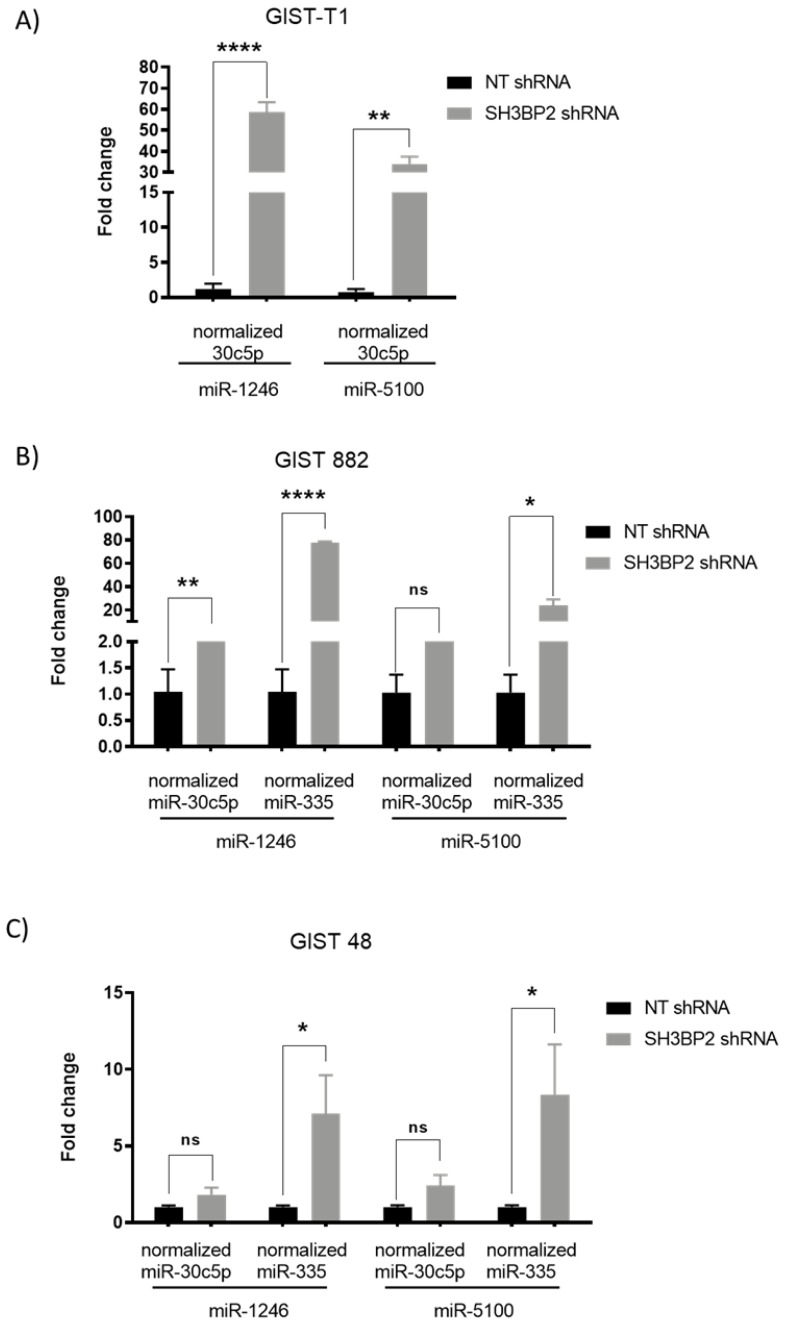
Validation of miRNA upregulation after SH3BP2 silencing by Real-Time PCR in GIST cell lines. GIST-T1, GIST882, and GIST48 cells were transduced with a non-target shRNA sequence or a specific shRNA SH3BP2. MiR-335 and miR-30c5p were used as housekeeping miRNAs. (**A**) GIST-T1 Data represent one biological replicate performed two times. (**B**) GIST 882 Data are representative of two biological replicates performed two times. (**C**) GIST 48 Data represent three biological replicates performed two times. (* *p* < 0.05, ** *p* < 0.01, **** *p* < 0.0001; Unpaired *t*-test; mean ± SEM).

**Figure 4 cancers-14-06198-f004:**
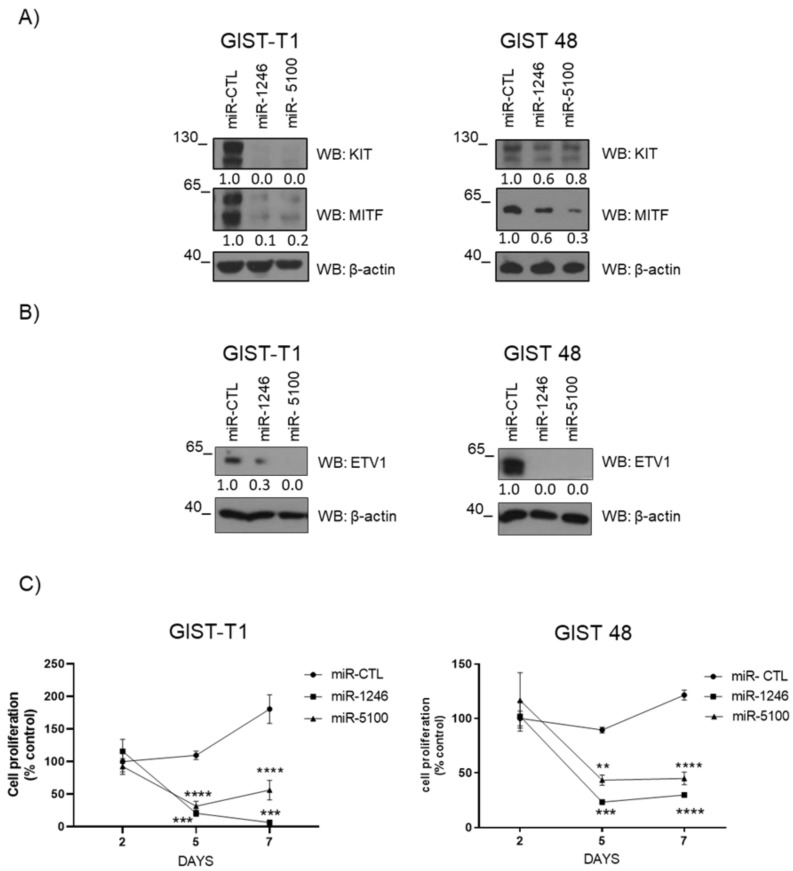
MiR-1246 and miR-5100 reduce cell proliferation in GIST cells. Western blot was performed on the 5th day after lentiviral transduction in GIST-T1 and GIST 48 to determine levels of (**A**) KIT and MITF, (**B**) ETV1. β-actin was used as load control. (**C**) Cell proliferation assay was performed by WST-1 on the 2nd, 5th, and 7th days after lentiviral transduction. (** *p* < 0.01, *** *p* < 0.001, **** *p* < 0.0001; one-way-ANOVA with Bonferroni’s post-hoc test) n = 3. GFP-miR-CTL was used as a control.

**Figure 5 cancers-14-06198-f005:**
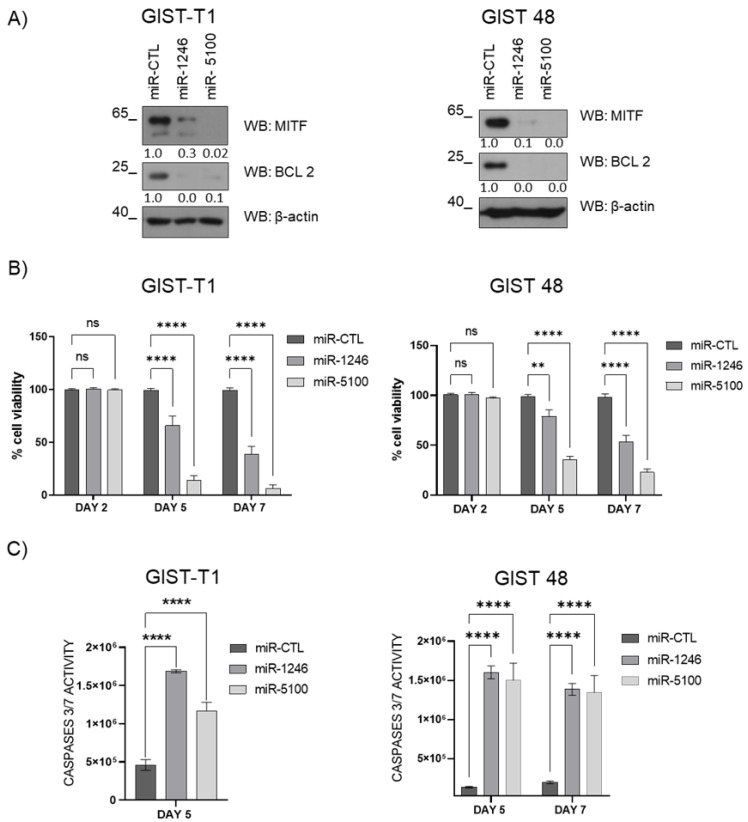
MiR-1246 and miR-5100 induce apoptosis in GIST cells. (**A**) Western blot was carried out on the 5th day after lentiviral transduction in GIST. MITF and BCL2 levels were assessed; β-actin was used as load control. (**B**) Viability was evaluated by crystal violet on the 2nd, 5th, and 7th days after lentiviral transduction. Statistical significance (** *p* < 0.01, **** *p* < 0.0001; one-way-ANOVA with Bonferroni’s post-hoc test) GIST-T1 n = 4; GIST 48 n = 3. (**C**) Caspase 3/7 activity was measured on the 5th day on GIST-T1; the 5th and 7th day on GIST 48 post lentiviral transduction (**** *p* < 0.0001; Unpaired *t*-test; mean ± SEM) n = 3. GFPmiR-CTL was used as a control.

**Figure 6 cancers-14-06198-f006:**
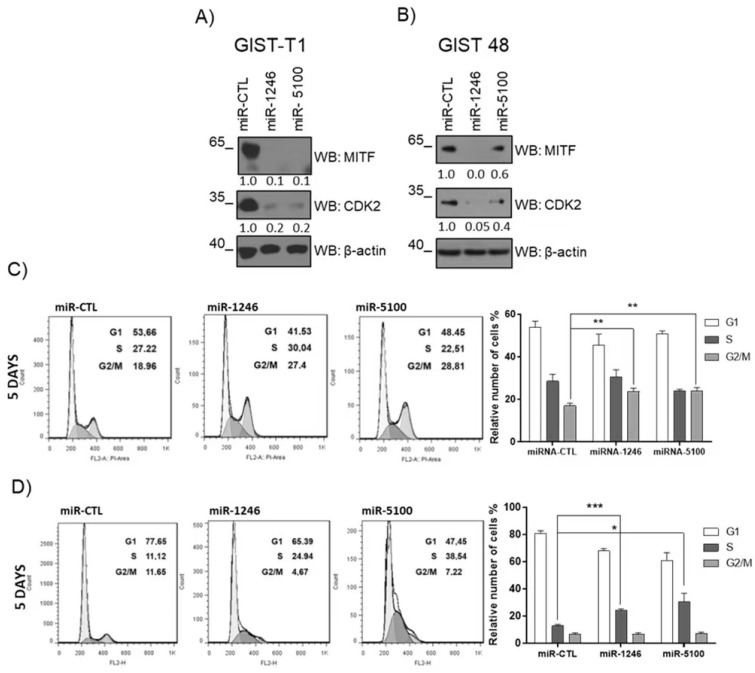
MiR-1246 and miR-5100 arrest cell cycle in GISTs cells. Western blots were performed on GIST cells on the 5th days after lentiviral miRNAs transduction; lysates were analyzed to determine MITF and CDK2 levels in (**A**) GIST-T1 and (**B**) GIST 48. Cell cycle assay was performed by FACS, and miRNA–CTL was used as a control. Results were analyzed by Dean/Jett/Fox model Flow jo 7.0 software (**C**) GIST-T1 n = 4, (**D**) GIST 48 n = 3 (* *p* < 0.05, ** *p* < 0.01, *** *p* < 0.001; Unpaired *t*-test; mean ± SEM).

**Figure 7 cancers-14-06198-f007:**
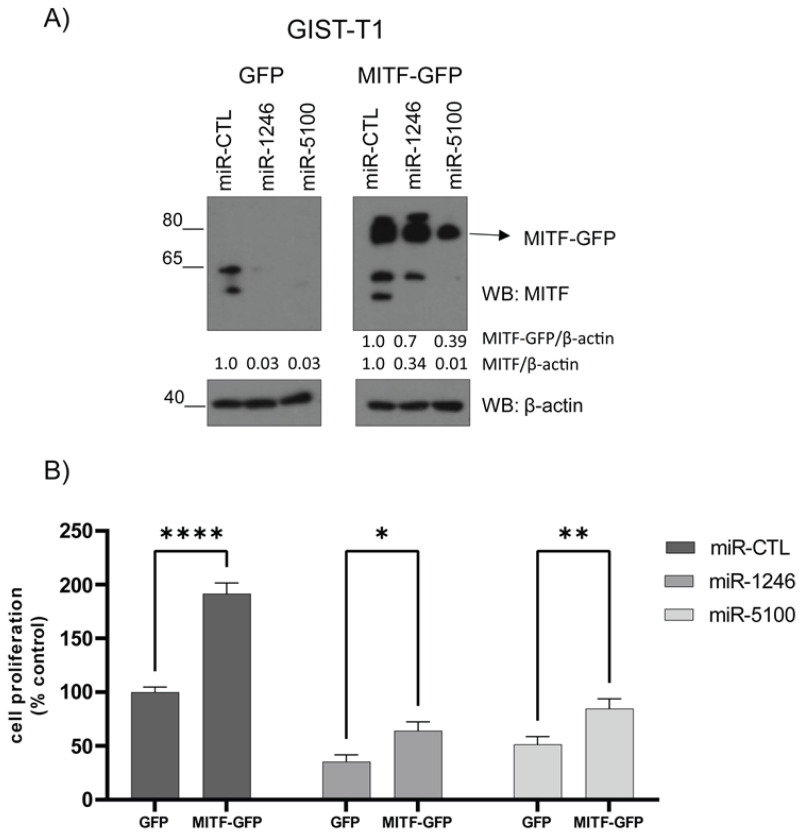
Overexpression of MITF reverses the phenotype produced by miRNAs in GIST-T1. Cells were transiently transduced with GFP or MITF-GFP plasmids on the 3rd day after lentiviral miRNA transduction. (**A**) Western blot shows MITF-GFP and MITF (endogenous) levels in GFP or MITF-GFP overexpressed cells. β-actin was used as a control. (**B**) Cell proliferation was measured by WST-1 on the 7th day after lentiviral miRNAs transduction. (* *p* < 0.05, ** *p* < 0.01, **** *p* < 0.0001. One-way ANOVA with Bonferroni’s post-hoc test) n = 3.

## Data Availability

The datasets used and analyzed during the current study are available in the article and Appendix A or from the corresponding author at reasonable request. GSE213777: Accession number for SH3BP2-silenced GIST microarray.

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
