# Peer review of "SH3BP2 Silencing Increases miRNAs Targeting ETV1 and Microphthalmia-Associated Transcription Factor, Decreasing the Proliferation of Gastrointestinal Stromal Tumors"

_cancers, 2022, doi:10.3390/cancers14246198_

Round 1

Reviewer 1 Report

Proaño-Pérez et al conducted a series of in vitro assays in gastrointestinal stromal tumors (GISTs) cell lines to investigate regulation between SH3BP2, ETV1, MITF, and two microRNAs (miR1246 and miR5100). Overall, this study is good to provide potential molecular mechanisms underlying GSITs. The authors identified these two miRNAs based on miRNA microarray in silenced SH3BP2 GIST cell lines and that they might target ETV1 and MITF from multiple miRNA databases. These two miRNAs affected cell behaviors when they were overexpressed in GIST cells. Meanwhile, they observed reduced protein abundance of ETV1 and MITF in the above cell lines. Did authors examine cell phenotypes by promoting protein abundance of ETV1 and MITF in cell lines with silenced miRNAs? That would provide solid evidence for the regulation between these two miRNAs and ETV1 and MITF. Besides, I still have minor comments below:

1. Line 182, the authors need to describe “ETV1 expression” clearly because “expression” usually refers to gene expression. It could confuse readers. “level” or “abundance” would be better for “protein”. Also please check this throughout the manuscript.

2. Please carefully check gene name “SH3BP2” is correct in all figures.

3. In Figure 1, why were two protein signals detected for ETV1 and MITF in cell line GIST-T1?

3. Line 204, add a period after “cell type”

4. Lines 210 and 214, “(p-value <0.5)” should be 0.05?

6. The gene names in the gene context should be in Italic font.  

7. It will be better if the authors add the RT-PCR results of the other three miRNAs to supplementary data.

8. Line 221,  What was the purpose of this analysis “The database miRNANet was used to predict miRNA-target interactions with these miRNAs.”? And what was the result here? I saw other databases used in line 226.

Reviewer 2 Report

In this study Proano-Perez et al. studied the molecular mechanisms involved in SH3BP2 silencing and decreased cell viability in GIST cell lines.

Comments:

The data provided to support mir-1246 and mir-5100 regulation by SH3BP2 in GIST-T1 cells are not convincing because: 1) no miRNA microarray was done for GIST-T1 cell line and 2) Real-time PCR validation data for GIST-T1 are representative of only one biological replicate, normalized by a single housekeeping.

This undermines the study’s value, because the subsequent experiments (paragraph 3.7: demonstration that MITF overexpression restores cell proliferation) and (paragraph 3.8:  evaluating the effect of LNA treatment on apoptotic phenotype in SH3BP2 silenced cells) were done only in this one (GIST-T1) cell line.

Reviewer 3 Report

KIT mutation is oncogenic drive in the majority of GIST and serves as key therapeutic target. SH3BP2 positively regulates expression of KIT and downstream MITF and ETV1. Conversely, MITF and ETV1 also enhance KIT expression. In this study, Proaño-Pérez et al. performed miRNA profiling in imatinib-sensitive and resistant GIST and found miR-1246 and miR-5100 were generally induced in SH3BP2 depleted GIST cells. miR-1246 and miR-5100 negatively regulated protein expression of MITF, ETV1 and KIT, which probably resulted in reduced cell viability, increased apoptosis and redistributed cell cycle.

This is an interesting study. Manuscript is written well. Current data is a little insufficient but still supports major conclusions. Below are my concerns and suggestions that may help improve the quality of this manuscript.

1. It is unknown if SH3BP2 directly suppresses expression of miR-1246 and miR-5100 or not. Does SH3BP2 skip KIT to regulate miR-1246 and miR-5100 because miRNAs were down in imatinib-sensitive and resistant GIST cells?

2. It is hasty to conclude that “miR-1246 and miR-5100 target ETV and MITF”. In supplementary Fig 1 authors only showed putative response elements of miR-1246 and miR-5100 on MITF transcript. Such elements on ETV1 were not shown. Then authors just showed downregulation of MITF and ETV1 after expression of two miRNAs. Standard experiments for miRNA target validation were missing: (1) Luciferase reporter assay to show miRNA mimics reduce while miRNA antisenses (or LNAs specific for miRNAs) increase luciferase activity of reporter constructs with WT binding regions but not mutant binding regions on transcripts of MITF and ETV1; (2) western blot to show accumulation of MITF and ETV1 protein after treatment of miRNA antisenses (or LNAs); (3) correlation of miR-1246 and/or miR-5100 with MITF and/or ETV1 protein in patient (database or detection in GIST patient samples).

3. Cell viability change in GIST cells expressing miRNA antisenses was missing.

4. Using two housekeeping miRNAs in Fig 3 is great. MiRNA expression in Fig 3A should also be normalized to miR-335 given that there was big difference of fold change between two normalization groups. Will miR-1290 significantly increase in shSH3BP2 cells when normalized to miR-335? Change SH3BP shRNA to SH3BP2 shRNA in Fig 3 to unify format.

5. In Fig 4A, both miRNAs significantly downregulated KIT in imatinib-sensitive GIST-T1 cells. In contrast, KIT downregulation was weak in imatinib-resistant GIST48 cells. It is interesting and authors should explain why. Because positive feedback loops of MIST-KIT and ETV1-KIT does not exist in GIST48 cells?

6. In Fig 4C, viability of miR-CTL GIST48 only increased by 20% within 7 days? If this cell line was not proliferative, how did authors expand it for so many experiments? In other studies (Fig 5 in https://doi.org/10.3390/cells9061333; Fig 4C in https://doi.org/10.1111/jcmm.13502), viability of GIST48 was doubled within 3-4 days. Another reason was that miR-CTL had cytotoxicity.

7. It is better to run 6 samples in Fig 7A in one gel for straightforward comparison. Moreover, plasmid dose was too high. Ectopic MITF level after expression of miR-1246 or miR-5100 was still much higher than endogenous MITF in miR-CTL sample. This was simply overexpression but not reconstitution. MITF overexpression enhanced cell proliferation and MITF-OE+miR-1246/5100 cells were naturally more proliferative than miR-1246 /5100 cells. Besides, does binding region of miR-5100 locate in coding region or 3’UTR of MITF cDNA? It was not illustrated in supplementary Fig 1. Did Ectopic MITF cDNA harbor binding region of miR-5100? Why miR-5100 reduced ectopic MITF level?

8. Supplementary Figure 3, “Fluorescence FAN LNA miR-CTL, miR-5100, and miR-1246 were measured by microscopy fluorescence 24h after LNA treatment.” But cell viability was performed 4 days after lentivirus infection. Were LNAs still in cells at that time? If so did these LNAs prevent RNA hybridization but still could not rescue apoptosis in shSH3BP2 cells. Authors should (1) perform RNA blot analysis to detect hybridization of miR-1246 and miR-5100 (U6 as a control) in LNA-treated shSH3BP2 cells; (2) measure cell viability again as supplementary Fig 3A and see if effective miRNA silencing rescues viability inhibition; (3) see if protein level of MITF and ETV1 are rescued by LNAs. These experiments will provide key evidences demonstrating miR-1246 and miR-5100 mediate viability inhibition upon SH3BP2 depletion in GIST cells and should be put in main figure.

9. Authors wrote in row 67 “herein, we aim to study ETV1 involvement in the pathway”. However, authors did not provide enough evidence linking ETV1 and SH3BP2-KIT-MITF signaling. Positive ETV1 regulation by SH3BP2 (foreseeable because of known KIT-MAPK-ETV1 axis), negative ETV1 regulation by miR-1246/5100 and negative regulation of miR-1246/5100 by SH3BP2 are only results readers can find here. Does miR-1246 and/or miR-5100 restriction by SH3BP2 contribute to ETV1 expression? What are positions of KIT and MITF in the SH3BP2-miR-ETV1 signaling? Does ETV1 downregulation after miRNA overexpression contribute to phenotypic change similar to MITF downregulation?

Round 2

Reviewer 2 Report

The revised manuscript is acceptable

Reviewer 3 Report

Authors addressed my concerns appropriately. I have no more comment. I feel manuscript can be accepted in current format.